# Chain-End Effects on Supramolecular Poly(ethylene glycol) Polymers

**DOI:** 10.3390/polym13142235

**Published:** 2021-07-07

**Authors:** Ana Brás, Ana Arizaga, Uxue Agirre, Marie Dorau, Judith Houston, Aurel Radulescu, Margarita Kruteva, Wim Pyckhout-Hintzen, Annette M. Schmidt

**Affiliations:** 1Institute of Physical Chemistry, University of Cologne, 50939 Cologne, Germany; ana.arizaga@gmail.com (A.A.); uagirre@smail.uni-koeln.de (U.A.); MarieDorau@gmx.net (M.D.); annette.schmidt@uni-koeln.de (A.M.S.); 2Jülich Centre for Neutron Science (JCNS-1) at Heinz Maier Leibnitz-Zentrum (MLZ), Forschungszentrum Jülich GmbH, 85748 Garching, Germany; judith.houston@esss.se (J.H.); a.radulescu@fz-juelich.de (A.R.); 3Jülich Centre for Neutron Science (JCNS-1), Forschungszentrum Jülich GmbH, 52428 Jülich, Germany; m.kruteva@fz-juelich.de (M.K.); w.pyckhout@fz-juelich.de (W.P.-H.)

**Keywords:** supramolecular polymer, thymine-1-acetic acid, diamino-triazine, 2-ureido-4[1H]-pyrimidinone, homocomplementary, heterocomplementary, Flory–Huggins interaction parameter, functionalization, hydrogen bonding

## Abstract

In this work we present a fundamental analysis based on small-angle scattering, linear rheology and differential scanning calorimetry (DSC) experiments of the role of different hydrogen bonding (H-bonding) types on the structure and dynamics of chain-end modified poly(ethylene glycol) (PEG) in bulk. As such bifunctional PEG with a molar mass below the entanglement mass $M_{e}$ is symmetrically end-functionalized with three different hydrogen bonding (H-bonding) groups: thymine-1-acetic acid (thy), diamino-triazine (dat) and 2-ureido-4[1H]-pyrimidinone (upy). A linear block copolymer structure and a Newtonian-like dynamics is observed for PEG-thy/dat while results for PEG-upy structure and dynamics reveal a sphere and a network-like behavior, respectively. These observations are concomitant with an increase of the Flory–Huggins interaction parameter from PEG-thy/dat to PEG-upy that is used to quantify the difference between the H-bonding types. The upy association into spherical clusters is established by the Percus–Yevick approximation that models the inter-particle structure factor for PEG-upy. Moreover, the viscosity study reveals for PEG-upy a shear thickening behavior interpreted in terms of the free path model and related to the time for PEG-upy to dissociate from the upy clusters, seen as virtual crosslinks of the formed network. Moreover, a second relaxation time of different nature is also obtained from the complex shear modulus measurements of PEG-upy by the inverse of the angular frequency where *G*’ and *G*’’ crosses from the network-like to glass-like transition relaxation time, which is related to the segmental friction of PEG-upy polymeric network strands. In fact, not only do PEG-thy/dat and PEG-upy have different viscoelastic properties, but the relaxation times found for PEG-upy are much slower than the ones for PEG-thy/dat. However, the activation energy related to the association dynamics is very similar for both PEG-thy/dat and PEG-upy. Concerning the segmental dynamics, the glass transition temperature obtained from both rheological and calorimetric analysis is similar and increases for PEG-upy while for PEG-thy/dat is almost independent of association behavior. Our results show how supramolecular PEG properties vary by modifying the H-bonding association type and changing the molecular Flory–Huggins interaction parameter, which can be further explored for possible applications.

## 1. Introduction

Non-covalent interactions and self-assembly are responsible for the organization of many biological systems [1,2,3,4]. Often, more than one type of non-covalent interaction plays a paramount role. A very well-known example is DNA, where hydrogen bonding (H-bonding) between the base pairs (thymine, adenine, guanine, cytosine) as well as π-π stacking and hydrophobic interactions give DNA its famous double helix structure [5,6,7,8]. Other examples are lipid bilayers and folding of proteins into helices or ß-sheet stabilized by non-covalent interactions, especially hydrogen bonds (H-bonds). Therefore, studying hydrogen bonding interactions allows a better understanding of many biological phenomena, such as the link between structure and function of proteins. In recent years, the highly directional physical interactions based on H-bonds have been applied in a fundamentally different way to form supramolecular polymers, as a means to mimic the biological self-assembly and organization [9,10,11,12,13,14,15,16,17,18,19,20,21]. Indeed, in these materials, modification of low molar mass polymers with functional groups that associate via H-bonding interactions give rise to a rich variety of self-organizing structures on the mesoscale with a multiplicity of macroscopic properties [22,23,24,25,26,27,28,29,30,31,32,33]. In particular, the nature of the H-bond interactions will largely influence the structure and dynamics of the supramolecular self-assembled structures. Their properties can therefore be widely tailored by these parameters [3,34,35,36]. In this context, a fundamental understanding of H-bond interactions and their influencing parameters is necessary to control the structure and dynamics of the resulting supramolecular polymers. One of the simplest examples of associating supramolecular polymers is expected with bifunctional polymers. Typical building blocks are linear polymer chains carrying end functionalized binding groups with hydrogen bonding motifs.

Due to its specific features, such as its water solubility or biocompatibility, PEG is an interesting polymer for various industrial and biomedical applications. While some of us [13,18] have previously investigated similar bifunctional PEG polymers end-functionalized with heterocomplementary associating thy and dat in the bulk, almost no fundamental studies exist on simple bifunctional PEG polymers end-functionalized with homocomplementary associating upy in the bulk [37,38]. On the one hand, though being one of the most important and extensively studied H-bonding group [39,40,41,42,43,44,45], since the first reports from the group of Meijer et al. [38,46], the focus is mostly on water solution due to the formation of supramolecular hydrogels that show promising properties for application in regenerative medicine because of their ability to adapt to the natural environment these materials are brought into [16,47,48,49]. Indeed, in solution PEG-upy based supramolecular hydrogels change from polymer micelles to increasingly ordered structures, like fibrils depending both on the concentration and on the functionality type of the PEG polymer [3,50,51,52,53,54]. On the other hand, comparably detailed studies on supramolecular PEG-upy melt association are scarce. Additionally, here primarily the mechanical properties are investigated since the focus of these investigations have been also on the applications, especially supramolecular self-healing materials [9,50,55,56]. For PEG-thy/dat, it is known [13,18], that linear association and simple Rouse dynamics prevail based on small-angle neutron scattering (SANS), pulsed field gradient nuclear magnetic resonance (PFG NMR) and viscosity measurements. Moreover, a direct microscopic quantitative determination of the bond breaking relaxation time, *τ*_b_ is obtained using neutron spin echo (NSE) spectroscopy [18]. While these results reveal many details about the dynamics of PEG-thy/dat, there is rather limited discussion on the role of the segmental dynamics, either studied by rheology and calorimetry in this polymer.

Therefore, in this work we aim to compare on a fundamental level the influence of different homocomplementary and heterocomplementary H-bonding association types, on the structure and dynamics of Poly(Ethylene glycol) (PEG) based supramolecular polymers. Homocomplementary hydrogen bonding denotes the association of two identical end-groups, whereas in the heterocomplementary case, two different end-groups bind together. The heterocomplementary H-bonding association consists of thymine-1-acetic acid (thy) and diamino-triazine (dat) forming triple H-bonds and the homocomplementary association involves 2-ureido-4[1H]-pyrimidinone (upy) forming quadruple bonds. In this context, we present a combined study of small angle scattering (SAS), linear rheology and differential scanning calorimetry (DSC), to unravel the differences on the microscopic structure, and consequently on the underlying rheological macroscopical mechanisms, segmental dynamics and glass transition regimes of bifunctional PEG end-functionalized with different H-bonding thy/dat and upy association types. By comparing the different association types, we present a consistent interpretation of the data and discuss the parameters that play a key role on the different structural and dynamical behavior of these supramolecular PEG. Ultimately this knowledge can influence the choice of the end-functional end-groups for future applications using PEG polymers.

## 2. Materials and Methods

### 2.1. Samples

Bifunctional PEG-thy, PEG-dat and PEG-upy are synthesized according to procedures in the literature [13,46]. The specific synthetic details of the polymer blocks and their characterization are presented in the Appendix A. Table 1 repeats, in short, the results. For all polymers ^1^H nuclear magnetic resonance (NMR) was used to determine the number average molar mass ($M_{n}$) and the functionalization degree (f) and size exclusion chromatography (SEC) was used to determine the polydispersity index (PDI). Equimolar mixtures (50:50) of bifunctional PEG-thy and PEG-dat compounds are prepared by solution blending in chloroform and dried under high vacuum conditions for two days.

### 2.2. Flory–Huggins Interaction Parameter χ

The Flory–Huggins interaction parameters of the supramolecular polymers can be estimated from the individual solubility parameters for thy, dat, upy and PEG using the Hildebrandt–Scott equation:(1)$$\chi_{i-j}=\frac{V_{i}}{RT}{\left(\delta_{i}-\delta_{j}\right)}^{2}$$

$\chi_{i-j}$ is the functional group–polymer interaction parameter, $V_{i}$ is the molar volume of the respective functional group, R is the gas constant, *T* is the absolute temperature, and $\delta_{i}$ and $\delta_{j}$ are the solubility parameters for the functional group and the polymer, respectively. The molar volume of the respective functional group, $V_{i}$, is calculated using the following densities, *d*_thy_ = 1.23 g/cm^3^, *d*_dat_ = 1.5 g/cm^3^ and *d*_upy_ = 1.70 g/cm^3^ at 298 K [21,34], and the respective molar masses, $M_{\mathrm{thy}}$ = 126 g·mol^−1^, $M_{\mathrm{dat}}$ = 111 g·mol^−1^ and $M_{\mathrm{upy}}$ = 154 g·mol^−1^. The values of the solubility parameters are mostly empirical estimates and only a limited number of experimental values are available, therefore slightly different values for the same molecule appear in the literature [23,57]. The solubility parameters values estimates are $\delta_{\mathrm{PEG}}=22.0$ (J^1/2^·cm^−3/2^), $\delta_{\mathrm{thy}}=27.5$ (J^1/2^·cm^−3/2^), $\delta_{\mathrm{dat}}=28.0$ (J^1/2^·cm^−3/2^) and $\delta_{\mathrm{upy}}=33.0$ (J^1/2^·cm^−3/2^) [23,58]. The Flory−Huggins parameters, χ, between PEG and thy and dat, can be estimated at T = 298 K in good accuracy using Equation (1) from the empirical solubility parameters to be ~1.25 and ~1.42, respectively. These values can be considerate almost negligible in comparison to the Flory–Huggins interaction parameters for PEG and upy estimated also using Equation (1) to be ~4.42. The higher the interaction parameter, the lower is the compatibility between the polymer and functional end-groups.

### 2.3. Nuclear Magnetic Resonance (NMR)

^1^H-NMR spectroscopy of the studied polymers is performed using a Bruker DPX 300 (300.13 MHz) at ambient temperature. As an internal standard, the solvent signal is used. The chemical shift is assigned in ppm.

### 2.4. Size Exclusion Chromatography (SEC)

SEC is performed with Tetrahydrofuran (THF) as the eluent on a SEC system from hs GmbH, Germany, with the following components: a pump (intelligent pump Al-12, Flow), a degasser (Gastorr AG-32, Flow), an autosampler (S5250, Sykam, Eresing, Germany), an RI detector (RI2012–A, Schambeck, Bad Honnef, Germany), a UV detector (S3245 UV/Vis-detector, Sykam, Eresing, Germany) and a column system (pre-column with 100 Å pore size and three columns of 10,000 Å, 1000 Å and 100 Å, respectively) from MZ Analysentechnik; Germany with MZ-Gel SD plus as the stationary phase. Polystyrene standards (Polymer Laboratories) in the molar mass range of $M_{n}$ = 925 g∙mol^−1^ − 1.98 × 10^6^ g∙mol^−1^ are used for the calibration of the system.

### 2.5. Small Angle Scattering

Small angle neutron scattering (SANS) experiments are carried out at the SANS diffractometer KWS2@FRM2, Munich, Germany. Absolute scattering intensities are measured over a scattering range from *Q* = 0.0047 Å^−1^ to 0.44 Å^−1^ using sample-to-detector distances of 2 m, 4 m, and 8 m and corresponding collimation lengths. The conversion to absolute scale intensities was done by means of the incoherent scattering of a 1 mm H_2_O sample and a 1.5 mm Plexiglas foil, respectively. The experimental two-dimensional data were corrected in standard way for background and empty cell scattering, detector sensitivity and radially averaged. Incoherent contributions were determined from the largest *Q*-range accessed. The corresponding neutron scattering length densities (SLD) of the components are calculated to be ${\mathrm{SLD}}_{\mathrm{PEG}}=6.39\times 10^{-7\;}{Å}^{-2}$, ${\mathrm{SLD}}_{\mathrm{thy}}=2.06\times 10^{-6\;}{Å}^{-2}$, ${\mathrm{SLD}}_{\mathrm{dat}}=2.73\times 10^{-6\;}{Å}^{-2}$, ${\mathrm{SLD}}_{\mathrm{thy}/\mathrm{dat}}=2.38\times 10^{-6\;}{Å}^{-2}$ and ${\mathrm{SLD}}_{\mathrm{upy}}=2.45\times 10^{-6\;}{Å}^{-2}$.

Small angle X-ray scattering (SAXS) data are measured at the GALAXI diffractometer based in the institute JCNS-2 at Forschungsszentrum Jülich [59]. The X-ray source utilizes a liquid metal jet target of a GaInSn alloy as the anode to which 70 keV electrons are sent. The resulting X-rays are monochromatized to allow only Ga K-α radiation of *E* = 9.243 keV photon energy to pass to obtain a wavelength *λ* = 1.34 Å. Two 4-segment slits which are separated by 4 m distance collimate the beam and confine the size to about 0.7 × 0.7 mm^2^. A 3rd slit reduces the scattering from the edges of the 2nd one. A sample-to-detector distance of 80 cm calibrated using Bragg reflections from silver behenate resulting in a *Q*-range 0.05–0.7 Å^−1^ is used. Absolute intensities in (cm^−1^) were obtained by the calibration with a secondary standard, consisting of a hexafluoro-ethylene-propylene copolymer (Dupont). The measured polymers are sealed in borosilicate capillaries of 2 mm nominal inner diameter and placed in the vacuum chamber at an experimental temperature of 333 K. Standard corrections for cell scattering and detector efficiency is performed. The corresponding X-ray scattering length densities of the components are calculated to be ${\mathrm{SLD}}_{\mathrm{PEG}}=1.36\times 10^{-5\;}{Å}^{-2}$, ${\mathrm{SLD}}_{\mathrm{thy}}=9.36\times 10^{-6\;}{Å}^{-2}$, ${\mathrm{SLD}}_{\mathrm{dat}}=9.34\times 10^{-6\;}{Å}^{-2}$, ${\mathrm{SLD}}_{\mathrm{thy}/\mathrm{dat}}=9.35\times 10^{-6\;}{Å}^{-2}$ and ${\mathrm{SLD}}_{\mathrm{upy}}=9.29\times 10^{-6\;}{Å}^{-2}$.

### 2.6. Rheology

Rheological measurements are performed using an AR-G2 rheometer (TA Instruments, New Castle, DE, USA) with a plate-plate geometry (20 mm) in which approx. 0.5 g of sample is used. The gap between the plates is 1 mm. Strain sweep experiments are performed at a frequency of 1 Hz in the strain regime 0.001 < γ < 0.5. Frequency sweep experiments in the range of 0.1–100 Hz were carried out in the linear viscoelastic regime. From these measurements, the frequency dependent storage modulus *G′*, and loss modulus *G″* of the supramolecular polymers is determined. Steady shear viscosities of the supramolecular polymers are measured in the shear rate range 0.01 s^−1^ with a temperature stability within 0.05 K.

### 2.7. Differential Scanning Calorimetry

Differential scanning calorimetry (DSC) measurements are performed in standard sealed aluminum containers between 193 K and 373 K at a heating rate of 10 K∙min^−1^ using a heat-flux calorimeter DSC-1 (Mettler Toledo) with an empty aluminum container as reference. The data of the second heating run are used for analysis.

## 3. Results and Discussion

Supramolecular polymers based on H-bonding end-groups exhibit different morphologies that will consequently have a different impact on the dynamical properties of these respective systems. Here, we want to understand the role of two different end-functionalized H-bonding association types on the structural and dynamic phenomena of bifunctional PEG polymer. Equimolar (50:50) mixture of PEG-thy/dat form heterocomplementary associations and PEG-upy form homocomplementary associations between the H-bonding groups. By comparing these different association types and presenting a consistent interpretation of the data we learn which of the parameters influencing the hydrogen bond associations play the key role on the different structural and dynamical behavior of supramolecular PEG. In the following the structure by small angle scattering and the dynamics by rheology and DSC of both PEG-thy/dat and PEG-upy are presented individually, followed by a comparison between the structure and dynamics of both supramolecular polymers. Here, the most distinct differences are put in evidence and discussed in terms of the influencing parameters. Moreover, the most important characteristics of the bifunctional PEG polymer block used as base for the supramolecular polymer synthesis is summarized. In this way, the real influence of the different association types can be better distinguished.

Scheme 1 illustrates the structure of both the equimolar bifunctional PEG-thy and PEG-dat heterocomplementary association and the PEG-upy homocomplementary association.

### 3.1. Reference System

The building blocks of our supramolecular polymers consist of PEG polymer with a molar mass of around 2000 g·mol^−1^ (Table 1). In order to understand the influence of the different two types of H-bonding groups end-functionalized on bifunctional supramolecular PEG, the structural and dynamical parameters of unfunctionalized PEG polymer in the melt have to be known and are taken as a reference for this work.

#### 3.1.1. Structure

The structural parameters of PEG polymer have been previously determined by a number of neutron scattering experimental data [60,61,62,63]. Therefore, and taking into account the theoretical definitions for a random walk model in an ideal Gaussian polymer chain in the melt, the radius of gyration, $R_{g}$ and the end-to-end distance, $R_{e}$ have the following relation:(2)$$R_{e}=\sqrt{6R_{g}^{2}}=\sqrt{Nl_{\mathrm{st}}^{2}}=b\sqrt{N_{K}}$$
where N is the number of repeating monomer units (see Table 1), $l_{\mathrm{st}}$ is the statistical segment length, b is the Kuhn length and $N_{K}$ is the number of Kuhn segments [63,64,65]. Since N is known (value in Table 1 results from $M_{n}=N\cdot M_{0}\;$ with $M_{0}$ = 44 g∙mol^−1^ being the molar mass of the PEG monomer) and $l_{\mathrm{st}}$ ($l_{\mathrm{st}}^{2}$ = 33.75 Å^2^) [60,63] and b (7.6 Å) are constants for this polymer [60,61,62,63], only $\;R_{g}=15.2\;Å$ and $R_{e}=37.4\;Å$ and $N_{K}=24.2$ are determined for our PEG polymer as indicated in Equation (2) above [66,67]. These parameters are used to check and confirm the theoretical model approaches employed to describe PEG-thy/dat and PEG-upy structures by scattering experiments in the following sections.

#### 3.1.2. Dynamics

It is known that for unentangled polymer chains with a molar mass below the entanglement molar mass ($M_{e,\mathrm{PEG}}\approx 2000\;g\cdot {\mathrm{mol}}^{-1}$) [62] as is our case (see $M_{n}$ in Table 1), one expects Rouse-like behavior that relates to the dynamics of Gaussian chains, which is determined by the balance of viscous and entropic forces. In this context, such polymers typically show a Newtonian behavior represented by a constant zero-shear viscosity $\eta_{0}$, at different shear rates when studied by rheology [67,68]. Therefore in the Rouse regime, both the segmental relaxation time $\tau_{s}$, as well as the expected Rouse time $\tau_{R}$, can be extracted using the zero-shear viscosity as shown by the following equations [13,60,69]:(3)$$\tau_{s}=\frac{12M_{n}\eta_{0}}{N_{A}N^{2}\rho \pi^{2}k_{B}T\;}$$
(4)$$\tau_{R}=N^{2}\tau_{s}$$
where ρ is the density of the polymer melt, $N_{A}$ the Avogadro number, $k_{B}$ the Boltzmann constant and $M_{n}$ and N are given in Table 1. At temperatures well above the glass transition temperature $T_{g}$, the chain dynamics can be well described by the Rouse model, and in this case, $\tau_{s}$ as given in Equation (3) is related to the Rouse time $\tau_{R}$, defined above in Equation (4) as the expected longest relaxation time for a chain with N repeating monomer units [13,60,69].

The steady-shear viscosity for our PEG polymer block is measured as studied by rheology in the temperature range from 333 K to 383 K, with 10 K as temperature interval and is shown in Table 2 for all the measured temperatures. Accordingly, it is possible to calculate the segmental relaxation time $\;\tau_{s}$ and Rouse time $\tau_{R}$, at the same temperatures for our PEG polymer block as from Equations (3) and (4), respectively that are also displayed in Table 2. These results of the relaxation times are an important input for the discussion of the segmental dynamics of the associated PEG supramolecular polymers and its temperature dependence is plotted below in Figure 4. In fact, the $\tau_{s}$ values are very similar to the segmental relaxation times taken from time of flight (ToF) experiment, a neutron scattering technique for a PEG of similar molar mass [69].

Additional information on the segmental mobility can be obtained by DSC measurements. Figure 1 displays the calorimetric curves for our PEG polymer block. The thermogram shows two transitions: the glass transition ($T_{g}$) and melting temperatures ($T_{m}$), with increasing temperature, respectively, though the location of $T_{g}$ is not clear. The value of $T_{g}$ in the thermogram is affected by the degree of crystallinity of the PEG polymer, and according to the literature, $T_{g}$ for PEG polymer ranges from 207 K to 245 K, depending on the degree of crystallinity [70]. Therefore, an estimation is done using the known empirical relationship $T_{m}=1.55\;T_{g}$ [71]. The analysis reveals $T_{m}$ = 332.2 K, which is within the values presented in literature [71] and the estimated glass transition, based on the melting temperature value taken from the analysis to Figure 1 is $\;T_{g}=214.3\;K$.

### 3.2. Heterocomplementary Association

For understanding the influence of the different association type on the structure and dynamics of bifunctional PEG polymer, the study of bifunctional PEG modified at the ends by functionalization with H-bonding thy/dat groups is essential. In this section, both the morphology and mobility of supramolecular PEG-thy/dat in the melt is discussed and revised in terms of the literature.

#### 3.2.1. Structure

The morphology of the equimolar (50:50) mixture of bifunctional PEG-thy/dat is investigated in the melt by SAXS. Figure 2 shows the SAXS results at *T* = 333 K and *T* = 353 K for fully hydrogenated PEG-thy/dat, i.e., intensity *I*(*Q*) in absolute units (cm^−1^), versus scattering vector *Q*. The contrast is given by the electron density difference of the PEG polymer block vs. the H-bonding end groups. The PEG-thy/dat SAXS data are corrected for background scattering by subtraction of the contribution of the PEG polymer SAXS data. The as such corrected intensities indicate a clear correlation hole peak typical for a block copolymer that is observed at intermediate *Q* at *Q** roughly ~0.16 Å^−1^ [72]. This corresponds approximately to a distance of 39 Å, which is approximately the end-to-end distance $R_{e}$ of a Gaussian chain of our PEG polymer block (see above). The height of the correlation peak is given by the number of correlated blocks, the scattering length densities, and the Flory–Huggins parameters [64,73]. At high *Q*, the curve shows the same typical *Q*^−2^ behavior of random walk chain statistics where the identity of the polymer block structure is lost. The *Q*^−2^ behavior characterizes the polymer chain structure in a melt and therefore rules out any more compact or segregated morphologies. At the lowest *Q*, a parasitic scattering contribution following a *Q*^−4^ power law is observed, but since no information can be obtained regarding the morphology, it is not taken into account and it is assigned to electron deficient voids or dust in the glass-sealed capillaries [74].

The observation of a block copolymer-like scattering signal on supramolecular polymers requires the use of random phase approximation (RPA) formalism for multicomponent systems as has already been shown in literature [13,23]. In this work we consider the binary system of a multiblock copolymer, as A block being the PEG polymer and B block representing both associating end-groups into a single effective block, basing on the identical solubility parameters. Thereby the system can be modelled as the general multiblock copolymer as$\;{\left(\mathrm{AB}\right)}_{X}$ copolymer with association degree *X* of diblock units, representing the number of aggregated building blocks $N_{\mathrm{agg}}$. This can be compared to the weight-averaged polymerization degree ⟨*N*_agg_⟩_w_ in polycondensation theory [13]. We have thus applied a full two-component RPA including all interactions *χ*_ij_ with i, j = A, B. With the number of monomers denoted as $N_{A}\;$and $N_{B}$ (polymer and compound end-groups, respectively), the volume fractions $\phi_{A}$, and $\phi_{B}$, specific monomeric volumes $v_{A}$, and $v_{B}$, form factors *P*_AA_(*Q*), *P*_BB_(*Q*) and *P*_AB_(*Q*) and the interaction parameters *χ*_AB_ polymer−end groups, the structure factor is now given as:(5)$$S_{RPA}=\frac{S_{\mathrm{AA}}^{0}S_{\mathrm{BB}}^{0}-S_{\mathrm{AB}}^{0^{2}}}{\left(S_{\mathrm{AA}}^{0}+S_{\mathrm{BB}}^{0}+2S_{\mathrm{AB}}^{0}\right)-2\frac{\chi_{\mathrm{AB}}}{\sqrt{\nu_{A}\nu_{B}}}\left(S_{\mathrm{AA}}^{0}S_{\mathrm{BB}}^{0}+S_{\mathrm{AB}}^{0^{2}}\right)}$$

The dimensionless parameter $a=\left(Q^{2}l_{\mathrm{st}}^{2}\right)/6$ such that $R_{g}^{2}=aN,\;$where *N* is the specified “block length” (equivalent to the number of repeating monomer units of the polymer, in the case of PEG or *N*_A_ and to five repeating PEG monomer units in the case of *N*_B_) and reflecting the random-walk statistics of the building blocks, with $l_{\mathrm{st}}$ the effective statistical segment length of the Gaussian subchains resulting from the approximation of including the end groups into the more flexible building blocks, is used to calculate the form factors mentioned above. The individual blocks are taken as monodisperse. More details on the RPA model can be found in the Appendix A and in [13,23].

As can be seen from Figure 2 the scattering curves for both temperatures are very close to each other and the correlation hole peak is slightly higher for *T* = 353 K in comparison with *T* = 333 K. In fact, this is not surprising as the peak depends not only on the $N_{\mathrm{agg}}$ but also on the χ-parameter. The black lines in Figure 2 displays the resulting fit to the SAXS data using the modified RPA model and the results are summarized in Table 3. Since the data are in absolute units (cm^−1^) the theoretically computed scattering length densities were fixed. In addition, the number of repeating monomer units of the polymer block as well as for the compound end-groups was kept constant. This leaves only the $N_{\mathrm{agg}}$, the Flory−Huggins interaction parameters $\chi_{\mathrm{AB}}$ and the effective statistical segment length $l_{\mathrm{st}}$ as refining parameters. The statistical segment length for PEG-thy/dat, $l_{\mathrm{st}}$ = 7.0 Å, is determined for both temperatures, being in fair agreement with what was previously reported for similar systems using SANS [60,62]. The fit also yields a number of aggregates $N_{\mathrm{agg}}$ ≈ 19 for *T* = 333 K and $N_{\mathrm{agg}}$ ≈ 16 for *T* = 353 K, either via thy/dat, thy/thy and/or dat/dat. Again, as the PEG-thy/dat here analyzed is a fully hydrogenated sample, the typical correlation peak of a block copolymer structure can only arise from the correlation between the end-groups and the PEG polymer in consequence of the Flory−Huggins interaction parameters $\chi_{\mathrm{AB}}$, as written above. In fact, contrary to a preceding work on a similar supramolecular polymer [13], here the interactions between the components are taken into account and are judged on the basis of the estimations for the solubility parameters. The interaction parameter between both blocks (H-bonding end-groups and PEG polymer block) is about 1.49 and 1.57 for *T* = 333 K and *T* = 353 K, respectively, showing a small temperature dependence, as expected. One has to bear in mind that the obtained values incorporate the small difference of densities and contrasts due to the temperature increase, which are held constant during the fit procedure to the data. Nevertheless, these values are close enough within the error bars to be considered almost indistinguishable. Indeed, the value of $\chi_{\mathrm{AB}}$ at *T* = 333 K is very comparable with the estimated ones from the mean-field-like solubility parameter approach above. Nonetheless, this value is considerably lower compared to the one found for an analogue Poly(propylene oxide) system, which has a less polar chain and thus less compatibility to the hydrogen-bonding groups [23]. In this respect, the derived interactions, quantified by the Flory−Huggins parameters, χ, between PEG and thy and dat, can be approximated in good accuracy to be irrelevant, as reported in a previous investigation [13].

As a first conclusion, the existence of heterocomplementary associated end groups with low Flory−Huggins parameters results in polymer chain-like aggregates, which behave Gaussian-like and show random-walk behavior in agreement with former neutron scattering experiments to a similar system [13,23]. Accordingly, the value of the number of aggregates for both *T* = 333 K and *T* = 353 K, shows a decrease with increasing temperature as expected for H-bonding association weakening, and corresponds to an associated chain with a molar mass well above the entanglement molar mass of PEG ($M_{e,\mathrm{PEG}}\approx 2000\;g\cdot {\mathrm{mol}}^{-1}$). However, it must be pointed out that the sensitivity of the peak intensity to the degree of aggregation is limited and it becomes virtually insignificant for a weight-averaged number of aggregates ⟨*N*_agg_⟩_w_ > 13–15 [13].

#### 3.2.2. Dynamics

The knowledge of the impact of the structure on the dynamics of PEG-thy/dat is necessary to understand the role of the H-bonding groups association on bifunctional PEG polymer. As the dynamics of heterocomplementary thy/dat association type is amply developed for PEG polymer [13,18], here we simply summarize the dynamic phenomena of PEG-thy/dat relevant for the understanding of the key differences between the two different association types on the PEG dynamics.

The dynamics of PEG-thy/dat are studied by rheology, in particular, strain sweep experiments and frequency sweep experiments in the temperature range between 333 K and 393 K with an interval of 20 K. Figure 3a,b show representative results for PEG-thy/dat for the shear viscosity at different shear rates and the complex shear modulus, i.e., the storage (*G′*) and loss (*G″*) moduli at different angular frequencies, respectively. The measured viscosity is independent on the shear rate for all temperatures as shown in Figure 3a. This is to be expected in the Newtonian regime [13,67,75]. Moreover, taking into account the oscillatory shear measurements in Figure 3b, no crossover is visible, it is evident that the slope of *G*″ is around 1, and the viscous behavior dominates. Again, the typical behavior of a Newtonian fluid is observed.

Table 4 lists the measured zero-shear viscosity $\eta_{0}$ averaged over different shear rates obtained at the different studied temperatures. It is observed that the viscosity decreases with increasing temperature. Moreover, the viscosity of PEG-thy/dat is always higher than the viscosity of unfunctionalized PEG polymer (see Table 2). Table 4 also displays the ratio between PEG-thy/dat and PEG polymer zero-shear viscosity, which is larger for lower temperature indicating a systematic de-association with increasing temperature, because otherwise these values would be constant. In this context, as the associated chains can be also described by the Rouse theory, as previously demonstrated in [13] and [18], the ratio between the viscosity values yield the weight-averaged number of aggregated building blocks $N_{\mathrm{agg}}{}_{w}$ [13].

As a consequence of the hydrogen bond interactions weakening, $N_{\mathrm{agg}}$ decreases with temperature. The obtained weight-averaged number of aggregated building blocks $N_{\mathrm{agg}}{}_{w}$ (~20 at *T* = 333 K and ~15 at *T* = 353 K) are in the agreement with previously results for a similar system and with the SAXS results above [13]. According to the Rouse model, the segmental relaxation time $\tau_{s,\mathrm{td}}$ for PEG-thy/dat are determined using also the measured zero-shear viscosities $\eta_{0}\;$in Table 4 with Equations (3) and (4) too. However, in this case, due to the association between the PEG-thy/dat chains, one has to consider that the number of repeating monomer units N of the PEG polymer building block (see Table 1) has to be multiplied by the weight-averaged number of aggregated building blocks $N_{\mathrm{agg}}{}_{w}$ to obtain the total number of monomers $N_{\mathrm{td}}$ for the associated chain, $N_{\mathrm{td}}=N\times N_{\mathrm{agg}}{}_{w}$. The same happens to obtain the associated polymer number average molar mass $M_{n,\mathrm{td}}=M_{n}\times N_{\mathrm{agg}}{}_{w}$, where the PEG number average molar mass $M_{n}$ has to be also multiplied by $N_{\mathrm{agg}}{}_{w}$.

The results for $\tau_{s,\mathrm{td}}$ are also displayed in Figure 4. It is clear that the segmental relaxation time $\tau_{s,\mathrm{td}}$ of the supramolecular associated system is the same as of the PEG polymer block. This relaxation time is usually ascribed to Brownian motion of chain segments, and since the supramolecular associated PEG is based on the PEG polymer block, the similarity of these relaxation times is not surprising. In fact, this is also in agreement to previously pulsed field gradient diffusion measurements [13] where it is shown that indeed telechelic end-groups do not significantly affect the monomeric friction coefficient.

Figure 4a presents the temperature dependence of the segmental relaxation time for both PEG polymer and PEG-thy/dat and the Rouse time for PEG polymer. Also in Figure 4, the literature values [69] for the segmental relaxation time, for a PEG polymer with similar molar mass as the one used in this work as well as the literature values for the bond breaking times $\tau_{b}$ [18], of a similar PEG-thy/dat obtained by neutron spin-echo spectroscopy (NSE) are represented for comparison purposes. The values of $\tau_{b,\mathrm{NSE}}$ represent the characteristic time scale of H-bond breaking in the PEG supramolecular compounds and interestingly show retardation due to loss of internal chain stresses upon breaking. As can be observed the experimental values for $\tau_{s}$ are in very good agreement with the respective literature values [18,69].

The data for the Rouse time $\tau_{R}$ for PEG is approximated to the Arrhenius law by the equation [76]:(6)$$\tau =\tau_{0}e^{E_{a}/RT}$$
where $\tau_{0}$ is the relaxation time at infinite temperature that for the case of a molecule is related to the time needed to move into some free space [77], R is the ideal gas constant and $E_{a}$ is the activation energy. According to the fit to the Rouse times for PEG polymer data using Equation (6), $E_{a}=30.6\;$kJ∙mol^−1^. By comparing the latter with previous results for a similar number averaged molar mass PEG ($E_{a}=27.4\;$kJ∙mol^−1^) [67], the values are indeed close. The activation energy that is reported in literature [18] for the bond breaking $\tau_{b}$ is $E_{a}=45\;$kJ∙mol^−1^ for a PEG-thy/dat similar system. Seemingly a small activation is necessary to break the bond [18].

Additional information on the segmental dynamics can be obtained by DSC measurements. Figure 5 displays the calorimetric curve for PEG-thy/dat, where two phase transitions are observed very similar to what is observed for PEG polymer in Figure 1, i.e., the glass transition $T_{g}$, and melting temperature $T_{m}$, with increasing temperature, though again the position of $T_{g}$ is not clear. The accuracy of the $T_{g}$ value is relatively low, since the glass transition region is very difficult to detect as a consequence of the high crystallinity of the PEG-thy/dat. Therefore, an estimation is done as for unfunctionalized PEG polymer block above using the known empirical relationship $T_{m}=1.55\;T_{g}$ [71]. The analysis of the thermogram reveal a $T_{m}$ = 314.2 K. The estimated glass transition, based on the melting temperature value taken from the analysis to Figure 5 is $\;T_{g}=202.7\;K$.

It is well accepted that segmental dynamics is cooperative in nature. Normally the temperature dependence of the relaxation time of the segmental relaxation in a logarithmic plot of the relaxation time as a function of the reciprocal of temperature shows a departure from linearity, i.e., it does not show a linear dependence and it is therefore approximated by the Vogel-Fulcher-Tammann-Hess (VFTH) law [76]:(7)$$\tau =\tau_{0}exp\left[B/\left(T-T_{0}\right)\right],\;T_{0}<T_{g}$$
where $\tau_{0}$ is seen as a microscopic quantity related to the time a molecule needs to move into some free space [76], *B* is a constant and $T_{0}$ is a temperature that usually is between 30 K and 70 K below $T_{g}$. In this context, the temperature dependence of the segmental relaxation time for both PEG polymer and PEG-thy/dat is described using the VFTH equation. The glass transition temperature $T_{g}$, for both PEG polymer and PEG-thy/dat is obtained by doing an extrapolation to *τ* = 100 s with the parameters obtained from the fit to the data of $\tau_{s}\;$ using Equation (3). The rheological *T*_g_ is then compared with the calorimetric $T_{g}$ in Figure 4b, for PEG polymer and PEG-thy/dat, respectively. The glass transition values obtained are $T_{g}$ = 206.6 K in comparison with 214.3 K and 202.7 K, respectively. The values are very close and in fair agreement with each other and with the values of the $T_{g}\;$ of PEG polymer in literature [70,71]. One has to bear in mind though that the viscoelastic measurements are performed well above the glass region and that PEG polymer and PEG-thy/dat are in semi-crystalline state below the melting temperature $T_{m}$. This turns the observation of a glass transition signal on the thermogram rather difficult. Indeed, both PEG polymer and PEG-thy/dat have crystallization degrees ≥40% (see further Table 9). Under these circumstances, the difference found between the values of the different techniques is still acceptable.

### 3.3. Homocomplementary Association

Besides the structure and dynamics of the heterocomplementary supramolecular PEG-thy/dat polymer, the homocomplementary supramolecular PEG-upy in the bulk is here in detail studied. As such the influence of different associative H-bonding functional end-groups on bifunctional PEG polymer properties can be identified. In the following the structure and dynamics of PEG-upy in the bulk is presented.

#### 3.3.1. Structure

The structure of PEG-upy is studied in the melt by both SAXS and SANS. Figure 6 shows the obtained data for fully hydrogenated PEG-upy at *T* = 333 K and *T* = 348 K. The contrast is given by the electron density difference or by the scattering length density difference between the associate upy end groups and the hydrogenous PEG polymer component, depending on the experiment source, either X-rays or neutrons, respectively. The PEG-upy scattering data is corrected for background scattering by subtraction of the contribution of the PEG polymer data at the same temperature. Qualitative observations from the as such corrected small angle scattering pattern show that on one hand the intermediate *Q*-range follows a power law, *I*(*Q*)∝ *Q*^−*P*^, with P ~ 4 and on the other hand an interaction peak can be spotted eventhough toward lower *Q* values the scattering pattern is filled with high intensity. Both observations suggest that the morphology of this system consists of interacting compact structures due to phase segregation. Indeed the strong association between upy end-groups is largely favored as the rather high value of $\chi_{\mathrm{PEG}/\mathrm{upy}}$ points out [20,78]. Therefore, these compact structures are composed by the upy end-groups segregated in clusters separated by the PEG polymer chains, similarly to what is observed in colloidal and micellar systems [27,79]. Normally these cluster-like structures are approximated to hard spherical aggregates and are thus treated with the Percus–Yevick model [64,80,81,82]. The model assumes a suspension of hard spheres with repulsive interaction. The potential is infinite when two spheres touch each other. In the lowest *Q*-range a strong decay of the intensity following approximately a *Q*^−3^ power law is also evident; however, no relevant information can be obtained at this *Q*-range regarding the morphology.

The scattering intensity per unit of volume of a spherically symmetric particles as in the Percus–Yevick hard sphere model is well known and generally is written as [65,83]:(8)$$I\left(Q\right)=\phi \Delta \rho^{2}VF{\left(Q,R_{c}\right)}^{2}S\left(Q,R_{d}\right)$$
where ϕ is the volume fraction of particles, $\Delta \rho^{2}$ the contrast factor between polymer and upy groups forming the spherical particles and V the volume. $P\left(Q,R_{c}\right)=F{\left(Q,R_{c}\right)}^{2}\;$is the form factor and $S\left(Q,R_{d}\right)$ the structure factor due to the contribution of the interactions between particles. Basically, the scattering intensity of the disordered particles can be attributed to the form factor of spheres with radius $R_{c}$ (the spherical clusters) and the structure factor of hard spheres (the Percus–Yevick correlation radius due to the interactions between spherical clusters) with a radius of $R_{c}$. We assume that the spherical particles are polydisperse. The detailed description of this model can be found in the Appendix A and also in [64,80,83].

The black lines in Figure 6 correspond to the fit to the data using the Percus–Yevick model. The fit parameters, including the volume fraction of spherical clusters (*ϕ*), the mean sphere cluster radius ($R_{c}$) and the Percus–Yevick correlation radius of clusters, ($R_{d}$) are summarized in Table 5. As displayed in Figure 6, a reasonable adjustment is obtained and, besides a slight difference at low *Q* values, both SAXS and SANS data are very similar for both measured temperatures. In fact, the values from $R_{d}$ and $R_{c}$ as $2R_{d}-2R_{c}=42\dot{A}$ and $40\dot{A}$ for both SAXS and SANS data, respectively at *T* = 333 K and $40\dot{A}$ for both SAXS and SAXS at *T* = 348 K correspond approximately to the end-to-end distance of the PEG polymer ($R_{e}=37.4\;Å)$. Moreover, the volume fractions in PEG-upy are relatively low and roughly agree with the molar fraction of upy groups in the supramolecular polymer ($M_{n,\mathrm{upy}}/M_{n,\mathrm{PEG}-\mathrm{upy}}=$0.066). This is in accordance with the fact that highly attractive interactions decrease the critical volume concentration that depends sensitively on changes in the interaction between the aggregates or polydispersity, which here is relatively low, ranging from ~11% to ~30%. Actually, this parameter is difficult to characterize as depends greatly on the high *Q* scattering, whose range is relatively limited, especially for the SAXS data [84]. Added to this, the aggregation number, $N_{\mathrm{upy}}$, defined as the number of end groups per spherical cluster, is an important parameter to characterize the aggregates. From the fit parameter $R_{c}$, $N_{\mathrm{upy}}$ can be calculated as $N_{\mathrm{upy}}=\frac{4\pi R_{c}^{3}\rho N_{A}}{3M_{n,\;\mathrm{PEG}-\mathrm{upy}}}$. Taking the density of the supramolecular polymer melt $\rho_{\mathrm{PEG}-\mathrm{upy}}$ = 1.2 g·cm^−3^ then $N_{\mathrm{upy}}$ is calculated to be between 25 (SAXS) to 35 (SANS) at *T* = 333 K and 31 (SAXS) to 35 (SANS) at *T* = 348 K. These results are reasonable, taking into account the high interaction parameter between upy groups and PEG in accordance with literature [46]. Also concerning the temperature dependence, it is only observed on the SAXS data and therefore the SANS data at *T* = 348 K is not shown. Actually, the influence of temperature seems to be mostly seen by an increase of the hard-sphere cluster volume fraction (Table 5). However, taking into account the error bars of the parameters, this difference cannot be considered, in agreement to what is obtained by SANS. In this context, an increase of temperature from *T* = 333 K to *T* = 348 K has a very little effect on the PEG-upy structure.

It is demonstrated that PEG-upy forms disordered spherical cluster aggregates as shown by the good agreement of the Percus–Yevick model to the data analysis.

#### 3.3.2. Dynamics

The macroscopic dynamics of PEG-upy is studied by the characterization of the viscoelastic properties measured by rheology in the temperature range from 333 K to 393 K with interval of 10 K. Figure 7a,b present the viscosity measurements and the complex shear modulus, i.e., the storage (*G′*) and loss (*G″*) moduli at different angular frequencies, respectively.

Under steady shear, PEG-upy exhibits a shear thickening behavior characterized by an increase in effective viscosity when the shear rate increases past a certain critical value as seen in Figure 7a. Although rarely observed in common polymer melts or solutions, shear thickening effects have been observed in complex fluids including dense suspensions, wormlike micelles, and associating polymer solutions [85,86,87]. The shear thickening seems to be caused by shear-induced structural changes in all of these systems. Several theoretical models have been proposed to describe the shear thickening behavior, especially in associating polymer solutions [87]. Marrucci et al. [88] explored the possibility of shear thickening as arising due to a non-Gaussian chain stretching effect. On the basis of the Tanaka and Edward transient network model, they argued that under flow conditions polymer chains may elongate considerably, well into the non-Gaussian regime, as a result of chain stretching. In an extension, the free path model by Marrucci et al. further assumed that when the chain end dissociates from a network junction, it can only partially relax its extended conformation since it is soon recaptured by the network again. As a consequence, the maximum in the viscosity occurs at a critical shear rate when the ratio of the detachment frequency to the shear rate (which decreases monotonically with increasing shear rate) has not dropped to its asymptotic lower bound, and yet the polymer chains are already stretched close to their maximum extension. In the free path model, the critical shear rate (shear rate at the maximum viscosity) ${\dot{\gamma }}_{\max}$ is then estimated assuming that the elastically active chains reach full extension at the onset of shear thickening. The mean-square distance of the polymer chain is then proportional to the number of Kuhn segments, $N_{K}$ and the critical shear rate is approximately:(9)$$({\dot{\gamma }}_{\max})\;\approx N_{K}^{1/2}/\tau_{d}$$
where $\tau_{d}$ is the network relaxation time. The upy groups that aggregate as spherical clusters are the chain junctions and act as the network crosslinks, eventually forcing the chain extension in case of detachment. The relaxation time of the PEG-upy network $\tau_{d}$ is determined using ${\dot{\gamma }}_{\max}$ taken from Figure 7a. According to this picture, $\tau_{d}$ is correlated with the detachment relaxation time of the PEG chains from the network junctions, i.e., from the upy clusters.

Moreover, from the oscillatory shear measurements in Figure 7b it is shown that the storage modulus *G*′ dominates over the loss modulus *G*″ almost over all the studied frequency range. The slope of around 0.6 ($G^{\prime \prime }\approx \omega^{0.6}$), is lower than 1 ($G^{\prime \prime }\approx \omega$), found for Newtonian systems or polymer melts at flow and typical of network-like systems. The elastic behavior dominates as *G*’ is relatively constant pointing up to a constant plateau at around 10^4^ Pa lower than the plateau module $G_{N}^{0\;}\sim \;1.45\times 10^{6}$ Pa for PEG polymer [60]. According to simple rubber elasticity theory, the plateau modulus ($G_{N}^{0}$) for a supramolecular network is given by [89]:(10)$$G_{N}^{0}=\nu_{e,\mathrm{RE}}k_{B}T=\frac{f_{e}{\;N}_{A}k_{B}T\;\rho }{M_{n,\mathrm{PEG}-\mathrm{upy}}}$$
where $\nu_{e,\mathrm{RE}}$ is the number density of elastically active strands, $k_{B}$ is the Boltzmann constant, *T* is the absolute temperature, ρ is the density of the supramolecular polymer melt (1.2 g·cm^−3^), N_A_ is Avogadro constant, $M_{n,\mathrm{PEG}-\mathrm{upy}}$ is the polymer molar mass and $f_{e}$ is the fraction of bridging or elastically effective molecules. Considering a perfect network where all PEG polymer chains would adopt a bridging conformation between the upy clusters and thus all would be elastically effective, $f_{e}$ = 1 and $\nu_{e,\mathrm{RE}}\approx 512$ mol·m^−3^. In this case, the estimated plateau module $G_{N}^{0\;}\sim \;1.45\times 10^{6}$ Pa for PEG polymer is retrieved [60]. While simple rubber elasticity theory overestimates $G_{N}^{0\;}$ of supramolecular PEG-upy, the overestimation is not surprising. In fact, a lower value and an overestimation of the plateau modulus is found in literature [89,90], for systems that also form cluster-like structures due to H-bonding interactions [90]. Even though H-bonding can create crosslinks that built the network it also introduces defects on the polymeric strands that is formed too due to loops existence, lowering $G_{N}^{0}$. Indeed, the possibility of PEG chains to form loops within the same cluster instead of bridges to different clusters has to be considered and can explain the lower $f_{e}$ (see further in the text). Besides that, probable contributions from PEG-upy functionalized at only one chain end, which stem from incomplete reaction steps during the functionalization of the polymeric product with subsequent different propensities, inevitably contribute to a lower fraction of effective elastically molecules between the upy crosslinks. In this context, the lower value of $G_{N}^{0\;}$ found for PEG-upy can be rationalized to a bridging fraction $f_{e}$ close to 1%. Definitely the observation of a plateau modulus clearly shows that the liquid-like viscoelastic properties of PEG at the studied molar mass have changed into a network-like due to the presence of the upy groups, in accordance with what is observed previously by small angle scattering.

Additionally, a crossover ($G^{\prime \prime }=G^{\prime }$) between *G*′ and *G*″ is visible at higher frequencies, corresponding to the relaxation time (τc=1ωc) related to the segmental friction of the polymeric network strands, $\tau_{c}$. It can be defined as the equilibration time or the longest relaxation time of the chains trapped in the network due to the clusters that act as crosslinks.

Figure 8a plots the corresponding temperature dependence of both the network detachment relaxation time $\tau_{d}$ as taken from the viscosity measurements analysis, and the network longest relaxation time $\tau_{c}$, as obtained from the crossover between storage modulus *G′* and loss modulus *G″*.

The data for the network relaxation time $\tau_{d}$ is described using the Arrhenius law [77]. According to the fit, $E_{a,\;\tau_{d}}=43.3$ kJ/mol and the prefactor for the chain breaking time is 21 μs, which is well in the order of what would be expected for an attempt angular frequency (2 × 10^−5^ rad·s^−1^). The value of $E_{a,\;\tau_{d}}$ is also as expected within the typical bond energies for H-bonding (10–65 kJ·mol^−1^) [30,38] known in literature. According to the free path model, the activation energy of the association is related to the viscosity maximum by $\eta_{\max}\approx \nu_{e,\mathrm{FP}}E_{a}\tau_{d}=\left(E_{a}/k_{B}T\right)\eta_{0}$, where $\eta_{\max}$ is the viscosity at ${\dot{\gamma }}_{\max}$, $\nu_{e,\mathrm{FP}}$ is the molar density of elastically active strands as defined in Equation (10) and $\eta_{0}$ is the zero-shear viscosity. According to the free path model assumptions, the zero-shear viscosity, $\eta_{0}$ can be calculated, as well as the molar density of elastically active strands $\nu_{e,\;\mathrm{FP}}$, since from the analysis above,$\;\tau_{d}$, $E_{a,\;\tau_{d}}$ are known and $\eta_{\max}$ can be directly extracted from Figure 7a.

Table 6 summarizes the viscosity maximum $\eta_{\max}$, the plateau modulus $G_{N}^{0\;}$, the molar density of elastically active chains $\nu_{e}$ from both theories and the zero-shear viscosity $\eta_{0}$ at the measured temperatures. The obtained zero-shear viscosity $\eta_{0}$ values seem to agree to the experimental data values at very low shear rates $\dot{\gamma }$ (Figure 7a). The molar density of elastically active strands $\nu_{e,\mathrm{FP}}$, is compared to $\nu_{e,\mathrm{RE}}$ calculated using Equation (10), and the values are quite compatible. The molar density of elastically active strands $\nu_{e}$ can also be obtained by small angle scattering. According to the structural picture it can be assumed that the molar density related to the Percus–Yevick correlation radius $R_{d}$, between clusters is given by $\nu_{e}=\frac{3\left(1-\phi \right)}{4\pi R_{d}^{3}N_{A}}$, and the molar density of elastically active PEG strands correspond to a value between 3.1 mol·m^−3^ (SANS) and 3.6 mol·m^−3^ (SAXS) at T=333 K and 3.1 mol·m^−3^ (SANS) and 3.2 mol·m^−3^ (SAXS) at T=348 K. Undeniably, these values are in very good agreement to the values calculated using the free path model assumptions on the rheology data. Moreover, from Table 6 we can also get the Weissenberg number Wi (shear rate multiplied with the relaxation time), at the shear rate at which the viscosity reaches its maximum point ($Wi={\dot{\gamma }}_{\max}\tau_{d}$). The Weissenberg number indicates the degree of orientation generated by the deformation. The Weissenberg number for our system is  ≈5, corresponding to a very high orientation of the polymer chains, which is also in conformity with the free path model.

All the above-mentioned arguments show the applicability of the free path model to understand the dynamics of PEG-upy and the good compatibility to the structural analysis.

Finally, information on the segmental dynamics of the network strands is being discussed. The temperature dependence of the longest relaxation time $\tau_{c}$, related to the segmental friction of the polymeric network strands for PEG-upy as known by polymer theory [91] is described using the VFTH equation as displayed in Figure 8b. The glass transition temperature $T_{g}$, for PEG-upy is obtained by doing an extrapolation to *τ* = 100 s with the parameters obtained from the fit to the data of $\tau_{c}\;$using Equation (7). The glass transition temperature $T_{g},\;$of PEG-upy polymeric network strands can be also obtained by DSC measurements. Figure 9 displays the calorimetric curve obtained for PEG-upy. As observed before for PEG polymer and for PEG-thy/dat, here also two-phase transitions are observed, i.e., the glass transition $T_{g}$, and melting temperature $T_{m}$, with increasing temperatures, respectively. Though the value of the glass transition temperature here is also influenced by the crystallinity degree, it gives a clear signal as depicted in Figure 9. Indeed, the crystallization degree is lowest of all the three studied polymers of around 30%. The analysis of the thermogram thus revealed $T_{g}$ ≈ 234.2 K and $T_{m}$ = 306.5 K and it is summarized further in Table 9. The extrapolation of the temperature dependence of network longest relaxation time $\tau_{c}$ for the low temperature dynamic regime gives a rheological value of $T_{g}$ = 245.6 K, which as expected, is closer to the results from the DSC ($T_{g}$ ≈ 234.2 K) than from the estimation, even though the $T_{g}$ from viscoelastic measurements is well above the glass region. Interestingly, an estimation done using the known empirical relationship $T_{m}=1.55\;T_{g}$, correspond to $T_{g}$ ≈ 197.7 K of the equivalent linear polymer [71]. This fact seems to point that the network-like structure and the phase segregation interfere with what is theoretically expected for the segmental dynamics of PEG-upy chains, taking only the $T_{m}$ and the lower crystallinity degree into account.

### 3.4. Comparison

The understanding of the role of the different H-bonding association types on the structure and dynamics of PEG polymer is essential for a fundamental view on the underlying mechanisms triggering the different properties of supramolecular polymers, in particular PEG based polymers. Therefore, a discussion of the key differences between the heterocomplementary and homocomplementary associative type on PEG polymer properties is essential.

#### 3.4.1. Structure

SAS results have shown that PEG-thy/dat associates into linear structures, proofed qualitatively by the *Q*^−2^ dependence at high *Q* and by the accurate description of the data by the RPA model, while PEG-upy presents a more compact phase segregated structure, consisting of a suspension of hard spheres with repulsive interaction, visible by the *Q*^−4^ dependence at intermediate *Q*-range and quantitatively by the good fit with Percus- Yevick model to the SAS data. Scheme 2 summarizes the molecular picture obtained by small angle scattering on both PEG-thy/dat (heterocomplementary association) and PEG-upy (homocomplementary association) reported in the previous Section 3.2 and Section 3.3.

It is shown that while thy/dat end groups bind only to one another creating an associating linear chain, upy H- bonding end groups are segregated in spherical clusters containing several of these groups. Surrounding these groups, many PEG chains as loops are observed but only one or two PEG chains are in fact, stretched to the end-to-end distance to another spherical cluster defining the distance between the disordered upy spheres. This picture is based on the parameters from the Percus–Yevick approximation fit to the small scattering data for PEG-upy that match very well to the structural characteristics of PEG polymer.

In this context, the obtained results can be interpreted in terms of the Flory–Huggins parameter, and seemingly, the higher the Flory–Huggins parameter interaction, the higher is the probability for phase segregation or compact structures, as well as the probability to associate.

It is demonstrated that bifunctional PEG polymer end-functionalized with different H-bonding association type groups have therefore different morphologies, quantified by the different range of interaction parameters values.

#### 3.4.2. Dynamics

Table 7 presents the parameters obtained using the VFTH equation (Equation (7)) fit to the segmental relaxation time $\tau_{s}$ data for PEG and PEG-thy/dat, as well as the parameters obtained using the VFTH equation fit to the network longest relaxation time $\tau_{c}\;$for PEG-upy. Table 8 displays the parameters obtained using the Arrhenius equation (Equation (6)) fit to Rouse time $\tau_{R}$ data for PEG as well as the Arrhenius equation (Equation (6)) fit parameters to the network detachment relaxation time $\tau_{d}\;$for PEG-upy. Table 8 also shows the fit parameters to $\tau_{b}$ for supramolecular PEG-thy/dat taken from [18] using the same equation (Equation (6)). Both Table 7 and Table 8 summarize the results obtained by rheology as reported individually in the previous Section 3.2 and Section 3.3 above.

The relaxation time at infinite temperature, $\tau_{0}$ for the relaxation times for PEG-thy/dat is relatively comparable to PEG polymer relaxation times at infinite temperature, ranging from 10^−4^ to 10^−3^ ns. The same happens for the dissociation time, $\tau_{b,\mathrm{NSE}}$. This means that at infinite temperature the relaxation time tend to be similar to the unfunctionalized PEG polymer, which is the reference for this study. This is not the case for PEG-upy. Both the network longest relaxation time, $\tau_{c}$ and the network dissociation relaxation time $\tau_{d}$ have higher values for the fit parameter $\tau_{0}$ as compared to PEG polymer. These observations can be understood in view of the strongly phase separated cluster network-like structure of PEG-upy in comparison to long associated linear chain of PEG-thy/dat and linear unfunctionalized PEG. Not only the characteristic time scale for the dissociation of a PEG-upy chain from the clusters in these supramolecular compounds but also the dynamics in general of PEG-upy is slower due to the network formation and phase segregation.

Another important point to discuss is the activation energy relative to the chain dynamics, i.e., the Rouse time of PEG ($E_{a}=\sim 30\;\mathrm{kJ}\cdot {\mathrm{mol}}^{-1})$ in comparison to $\tau_{b,\mathrm{NSE}}$ for a similar PEG-thy/dat supramolecular system $\left(E_{a}=\sim 45\;\mathrm{kJ}\cdot {\mathrm{mol}}^{-1}\right)$ [18] and the network detachment relaxation time, $\tau_{c}$ ($E_{a}=\sim 43\;\mathrm{kJ}\cdot {\mathrm{mol}}^{-1})$ for PEG-upy. Here, we discuss the activation energy of a virtual dissociation time energy for PEG-thy/dat and PEG-upy. Interestingly, though an increase from PEG polymer is observed, as expected due to the H- bonding end groups, the values of the activation energy for both PEG-thy/dat [18] and PEG-upy are almost the same. Apparently a small activation is necessary to break the H-bond when end-functionalized to PEG polymer as compared to other H-bonding supramolecular polymers [27,32,92]. Assuming a good correspondence between the increments in free energy and activation energy we may infer Δ$E_{a}$ ≃ 10–20 kJ∙mol^−1^ per H-bond [93]. This seems to be in accordance to the typical energy barrier specifically imposed by the H-bonds (10–65 kJ∙mol^−1^) found in literature [30,38]. This fact suggests the good assignment of the relaxation times. A higher value of the activation would seem to indicate that both the H-bonding groups dynamics are involved but also the chain dynamics. Also, it seems to be consistent to the relatively small influence of temperature on the structure of both PEG-thy/dat and PEG-upy.

Finally, the impact of the different H-bonding association type on segmental dynamics of PEG polymer are compared and discussed. Until now much less is known in both experiment and theory about changes of segmental dynamics in associating polymers, except for the recent work in references [32,94,95].

Table 9 displays the values of the rheological glass transition $T_{g,\mathrm{rheo}}$, the calorimetric glass transition $T_{g,\mathrm{DSC}}$, the melting temperature, $T_{m}$, the melting enthalpy, $\Delta H_{m}$ and the crystallinity degree for PEG polymer, PEG-thy/dat and PEG-upy obtained from the previous analysis in the sections above. The crystallinity degree is defined as the quotient between the enthalpy at the melting point, $\Delta H_{m},$ of PEG polymer, PEG-thy/dat and PEG-upy and the melting enthalpy of PEG polymer with 100% crystallinity, $\Delta H_{m,\mathrm{PEG}\;100\%}=196.8$ J∙g^−1^ [96].

It is observed that the addition of H-bonding groups on bifunctional PEG polymer decreases $T_{m}$ but increases $T_{g}$ for PEG-upy, regarding both $T_{g}$ and $T_{m}$ of unfunctionalized PEG polymer. In fact, especially for the melting temperature and the crystallization degree, this decrease is concomitant with the change on the supramolecular polymers structure due to the increase of the Flory–Huggins parameter. Indeed, the phase segregation of the upy groups into spherical clusters hinders the crystallization of the stretched PEG chains in larger extent than the chain length increases as in the case of PEG-thy/dat. Essentially the association observed for PEG-thy/dat is mainly reflected by the slight decrease of the crystallinity degree and the melting temperature, since the glass transition changes insignificantly for PEG-thy/dat in comparison for PEG polymer. Indeed, it has been also found for linear PDMS functionalized with small H-bonding groups no shift of $T_{g}$ with increasing number of association [95]. Moreover, values obtained by calorimetry and rheology are quite compatible given the fact that the fits to the data used to find the rheological glass transition are done at temperatures much higher than $T_{g}$ and the calorimetric $T_{g}$ is highly influenced by the crystallinity.

## 4. Conclusions

We studied the impact of different H-bonding association types, PEG-thy/dat heterocomplementary association and PEG-upy homocomplementary association, on the structure and dynamics of PEG polymer with molar mass below the entanglement mass $M_{e}$, using small angle scattering, linear rheology and DSC.

The Flory–Huggins interaction parameter is used to quantify the difference between the two association types. The interaction parameter is almost 3× higher for the PEG-upy homocomplementary association than for the PEG-thy/dat heterocomplementary type, so that the interaction parameter for PEG-thy/dat can be considered unimportant in comparison to the high χ of PEG-upy. In this context, the structure of the PEG-thy/dat and PEG-upy are different and change from linear to spherical-like phase segregated cluster association, respectively. In fact, the linear association is confirmed for PEG-thy/dat from the *Q*-dependence of a multiblock random phase approximation (RPA) structure factor, while for PEG-upy, the Percus–Yevick approximation is effectively used to model the inter-particle structure factor of upy association into spheres.

Consequently, while the viscoelastic properties of PEG-thy/dat are Newtonian-liquid like, and can be understood using the Rouse model, PEG-UPy viscoelastic properties are network-like, and the viscosity shows a shear thickening behavior that is interpreted using Marrucci’s free path model. According to the free path model, a network detachment relaxation time, defined as $\tau_{d}$, is obtained from the maximum in the viscosity that occurs at a critical shear rate for PEG-upy due to shear thickening. Moreover, a network longest relaxation time $\tau_{c}$, which is related to segmental friction of the polymeric network strands is obtained by the inverse of the angular frequency where *G′* and *G″* crosses from the network-like to glass-like transition relaxation time by the complex shear modulus measurements of PEG-upy. Therefore, the dynamics of PEG-upy is characterized by two different relaxation times of different origin, one related to the dissociation times of the PEG-upy chains from the upy-rich clusters or $\tau_{d}$ and the second to the longest relaxation time linked to the network-like segmental chain dynamics or $\tau_{c}$.

Through the Rouse model and using the zero-shear viscosity $\eta_{0}$, the Rouse time $\tau_{R}$ for PEG and the segmental relaxation time values for both PEG ($\tau_{s})$ and PEG-thy/dat $\left(\tau_{s,\mathrm{td}}\right)$ are obtained. As expected, the segmental times for the unfunctionalized and for the thy/dat functionalized PEG polymer are almost the same.

The activation energy $E_{a}$ of both dissociation related relaxation times for PEG-thy/dat and PEG-upy, $\tau_{b,\mathrm{NSE}}$ or bond breaking time obtained by the analysis to NSE data for a similar PEG-thy/dat [18] and $\tau_{d}\;$ for PEG-upy is very close for both H-bonding association types. These $E_{a}$ are in agreement to the known H-bonding energy values, even though the dynamics of PEG-upy is always slower in comparison to the dynamics of PEG-thy/dat and PEG polymer, but consistent to the small temperature influence on both PEG-thy/dat and PEG-upy structure.

Albeit there is a rather limited discussion on the segmental dynamics in associating polymers, in this study also the relaxation time related to the segmental dynamics of PEG-thy/dat and PEG-upy is compared. The extrapolation to τ = 100 s of the obtained temperature dependence of, $\tau_{s,\mathrm{td}}$ and $\tau_{c\;}$, for PEG-thy/dat and PEG-upy, respectively, provides an estimate of the glass transition temperature $T_{g}$, which is mostly in good agreement with $T_{g}$ analysed from the DSC results, even though the $T_{g}$ from viscoelastic measurements is well above the glass region. Mostly the $T_{g}$ for PEG-upy is higher than for PEG polymer due to network formation and phase segregation due to the clusterization of upy. Moreover, as found for other highly hydrophilic polymer functionalized with H-bonding groups too, the $T_{g}$ for PEG-Thy/dat is essentially independent of association [32,94], which is only reflected in the decrease of the crystallinity degree and melting enthalpy in comparison to unmodified PEG polymer.

Finally, a fundamental study on the influence of different H-bonding types on the structure and dynamics of supramolecular PEG have shown the good agreement between the microscopic and macroscopic observation by, small angle scattering, rheology and calorimetry, respectively. Comparing the Flory–Huggins interaction parameter difference between the chain and the H-bonding groups eventually allows predictions for the resulting microscopic structural and macroscopic viscoelastic properties. In other words, the associating groups must be selected properly particularly for a specific polymer to ensure that the desired properties can be achieved accordingly.

## Data Availability

Not applicable.

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
