# Peer review of "Chain-End Effects on Supramolecular Poly(ethylene glycol) Polymers"

_polymers, 2021, doi:10.3390/polym13142235_

Round 1
Reviewer 1 Report
Bras and coworkers report the chain-end effects on hetero- and homo-complementary supramolecular PEO polymers. The manuscript is well-written and presents the importance of how the H-bonding interactions at the polymer chain ends affect the physical properties of PEO polymers. I would like to support its publication in Polymers after the authors address the following points.
- Since the chain-end functionality of the PEO polymers is only around 90%, do the impure polymers have an influence on the observed physical properties?
- The polymers were blended in chloroform. Can the authors comment on how the solvent polarity affects the properties of the resultant polymer blends? If the polymer blend is annealed at a certain temperature, would the glass transition temperature vary?
Author Response
Response to Reviewer 1 Comments
----------------------------------------------------------------------
Dear Referee,
Thank you very much for your revision and valuable comments to our manuscript. We hope that the answers to your interesting questions will be satisfying for you.
Yours sincerely,
Authors of manuscript polymers-1276978
Comments and Suggestions for Authors
Bras and coworkers report the chain-end effects on hetero- and homo-complementary supramolecular PEO polymers. The manuscript is well-written and presents the importance of how the H-bonding interactions at the polymer chain ends affect the physical properties of PEO polymers. I would like to support its publication in Polymers after the authors address the following points.
- Since the chain-end functionality of the PEO polymers is only around 90%, do the impure polymers have an influence on the observed physical properties?
Response 1. As we answer to reviewer 2 we are strongly convinced that the ‘impurity’ in the functionality will shift the distribution to somewhat lower moments. The chain stopping mechanism if a monofunctional is built-in randomly will: i) increase the fraction of shorter chains, ii) give rise to a lower number-averaged molecular weight and iii) reach of the ideal polydispersity = 2,0 as expected for a polycondensation reaction in the limit of infinite molecular weights artificially earlier. Scattering and rheology measures the weight-average of the distribution and is therefore to a certain extent less sensitive to changes towards the lower-molecular weight tail. Therefore no effects on the short time scales are expected. Within error bars there is no observable deviation in the aggregation behaviour if the functionality is ~94% instead of 90%.
- The polymers were blended in chloroform. Can the authors comment on how the solvent polarity affects the properties of the resultant polymer blends? If the polymer blend is annealed at a certain temperature, would the glass transition temperature vary?
Response 2. Blending the oligomers with the different functional end-groups in a polar solvent guarantees the best possible mixture. In an apolar solvent groups would already homo or heteroassociate and blends of non-equilibrium oligomer-mixtures might get frozen in from the beginning. Maintaining the un-associated state prevents this and allows a high mobility of the species until the last moment. We hold this as a positive aspect of reaching an (unbiased) equilibrium.
It is an interesting question whether the glass transition would differ upon annealing on supramolecular polymers and we consider this in further work. Taking a look in literature [Macromolecules 2008, 41, 17, 6419–6430; Macromol. Chem. Phys. 2006, 207, 1262-1271], for example poly-L-lactic acid (PLLA) annealed at certain temperature below the melt state induced the cold crystallization of the polymer, and by changing this crystallization temperature, indeed the resulting glass transition of the semi-crystallized state changes, but only very slightly (between 2 to 8 °C). Moreover, to obtain such observation both the annealing, sample preparation and the DSC/DRS experiments that followed are done with very specific temperature steps and temperature treatments that we didn’t do in this work. Therefore no definite answer can be given, though having in mind the mentioned literature we should expect very little changes here, if any.
Submission Date
11 June 2021
Date of this review
23 Jun 2021 14:58:44

Reviewer 2 Report
The manuscript entitled “A Chain-end effects on supramolecular poly(ethylene glycol) polymers” by Brás et al., describes the association behaviour and dynamics of three purposely synthesized poly(ethylene glycol) polymers end-capped with 2,4-diamino-1,3,5-triazine (dat), thymine-1-acetic acid (thy) and 2-ureido-4[1H]-pyrimidinone (upy) groups, respectively. The gist of the reported research is represented by the discussion of the key differences between the heterocomplementary and homocomplementary associative type on the PEG polymer properties.
The work was carefully planned and conducted, the results are properly presented and the manuscript is quite well written. The first two sets of polymers, based on the dat/thy combination, were mixed to give heteroassociate by using hydrogen bonding interactions between complementary end-groups, while the upy-functionalized PEG derivative forms homoassociate.
The studies of the dat/thy heteroassociate are previously published by some of the authors (i.e. refs' 7,16) so the novelty of this work mainly relies on results obtained for the upy-functionalized PEG derivative. Yet, the manuscript can be published after some corrections and improvements:
1) Authors are recommended to check the reference numbers in the manuscripts. Some of them do not appear to match, for example what the reference 7 has with the following sentence: A very well known example is DNA, where hydrogen bonding (H-bonding) between the base pairs (thymine, adenine, guanine, cytosine) as well as π-π stacking and hydrophobic interactions give DNA its famous double helix structure?;
2) line 863: should be "..PEG-thy/dat heterocomplementary type..." instead of "PEG-thy/dat homocomplementary type";
3) use "eq." instead of "Äq." and "DMSO-d6" instead "DMSO-d6" in the supporting information section;
4) it is common to use two and not three decimal points for NMR data in the supporting information section.
Author Response
Response to Reviewer 2 Comments
----------------------------------------------------------------------
Dear Referee,
Thank you very much for your revision and valuable comments which help us to improve
our manuscript. We revised it according to your suggestions.
We hope that the corrections made in our manuscript will be satisfying for you.
Yours sincerely,
Authors of manuscript polymers-1276978
Comments and Suggestions for Authors
The manuscript entitled “A Chain-end effects on supramolecular poly(ethylene glycol) polymers” by Brás et al., describes the association behaviour and dynamics of three purposely synthesized poly(ethylene glycol) polymers end-capped with 2,4-diamino-1,3,5-triazine (dat), thymine-1-acetic acid (thy) and 2-ureido-4[1H]-pyrimidinone (upy) groups, respectively. The gist of the reported research is represented by the discussion of the key differences between the heterocomplementary and homocomplementary associative type on the PEG polymer properties.
The work was carefully planned and conducted, the results are properly presented and the manuscript is quite well written. The first two sets of polymers, based on the dat/thy combination, were mixed to give heteroassociate by using hydrogen bonding interactions between complementary end-groups, while the upy-functionalized PEG derivative forms homoassociate.
The studies of the dat/thy heteroassociate are previously published by some of the authors (i.e. refs' 7,16) so the novelty of this work mainly relies on results obtained for the upy-functionalized PEG derivative. Yet, the manuscript can be published after some corrections and improvements:
Comment: We apologize for the obvious misunderstanding. The work on the PEO-thy/dat, which is described in the underlying publication differs from the indeed already published one in the use of only protonated oligomers of PEO whereas the former results based on a isotopic mixture of protonated and deuterated sequences and functional groups that were smeared over in the chain ends. Moreover, in the present work using X-rays as the probe, the contrast comes from the groups with the oligomers. The 90% functionality of the newly synthesized oligomers is almost identical to the former one (~92-94%) and therefore the results and aggregation into linear associates will be very comparable, despite the obvious expected chain-stopping if a monofunctional oligomers is built in. In fact, the agreement is within 5% and underlines the consistency of the approach. The mono-functionality will distort the size distribution mainly towards lower molecular weights. Scattering and rheology measures the weight-average of the distribution and is therefore to a certain extent less sensitive to changes towards the lower-molecular weight tail. The effective polydispersity, which ideally approaches two for a molecular weight, , is probably reached somewhat earlier. Finally another point that is now additional to the previously published work concerns the tentative to gain new insights on the segmental dynamics of PEO-thy/dat that until now hasn’t yet been addressed.
- Authors are recommended to check the reference numbers in the manuscripts. Some of them do not appear to match, for example what the reference 7 has with the following sentence: A very well known example is DNA, where hydrogen bonding (H-bonding) between the base pairs (thymine, adenine, guanine, cytosine) as well as π-π stacking and hydrophobic interactions give DNA its famous double helix structure?;
Response 1) We thank the referee for pointing out this misunderstanding and we apologize for it. Reference 7 is now reference 13 and is changed after the sentence “In recent years, the highly directional physical interactions based on H-bonds have been applied in a fundamentally different way to form supramolecular polymers, as a means to mimic the biological self-assembly and organization”, while after the sentence “A very well known example is DNA, where hydrogen bonding (H-bonding) between the base pairs (thymine, adenine, guanine, cytosine) as well as π-π stacking and hydrophobic interactions give DNA its famous double helix structure” two new references on the structure and dynamics of DNA ([7] M. Hammermann, N. Brun, K. V. Kienin, R. May, K. Tóth, J. Langowski, Biophys. J. 1998, 75, 3057–3063. [8] K. R. Peddireddy, M. Lee, Y. Zhou, S. Adalbert, S. Anderson, C. M. Schroeder, R. M. Robertson-Anderson, Soft Matter 2020, 16, 152–161) are now included.
- line 863: should be "..PEG-thy/dat heterocomplementary type..." instead of "PEG-thy/dat homocomplementary type";
Response 2) We thank the referee for pointing out this mistake and we apologize for it. We have now corrected the sentence (red in the revised manuscript).
- use "eq." instead of "Äq." and "DMSO-d6" instead "DMSO-d6" in the supporting information section;
Response 3) We thank the referee for bringing this point to our attention. We have now changed the words correspondingly (red in the revised supporting information section).
- it is common to use two and not three decimal points for NMR data in the supporting information section.
Response 4) We thank the referee for bringing this point to our attention and we apologize for the mistake, which is now corrected in the revised supporting information section (red colored text).
Submission Date
11 June 2021
Date of this review
19 Jun 2021 17:33:52
